# The Role of User-Generated Content in the Sustainable Development of Online Healthcare Communities: Exploring the Moderating Influence of Signals

Xiaodan Yu, Hongyang Wang * and Zhenjiao Chen

School of Information Technology and Management, University of International Business and Economics, Beijing 100029, China
* Correspondence: wonghy7@outlook.com

**Abstract:** Grounded in signaling theory, this study explores the influence of user-generated content (UGC) within online healthcare communities on patient purchasing behavior, with the overarching goal of advancing the development of online medical consultation services and contributing to the sustainable evolution of the online healthcare community. Leveraging publicly available data from the "Haodf.com", we construct an empirical model of online medical consultation purchases, integrating principles from signaling theory and trust theory. Our analysis scrutinizes the effects of various forms of UGC on patient purchasing behavior, alongside the moderating influence of associated signals. The results demonstrate that knowledge-sharing articles authored by doctors and patient ratings positively impact consultation service purchases, whereas public displays of doctors' past consultation records impede such transactions. Furthermore, external signals were found to moderate the relationship between UGC and consultation service purchases. The implications of these findings offer actionable insights for stakeholders invested in online healthcare communities.

**Keywords:** online healthcare; consultation purchase; signaling theory; trust; UGC; sustainable development





## 1. Introduction

User-generated content refers to original content created by users and showcased or provided to other users through online platforms [1,2]. In online healthcare communities, this information may encompass health knowledge, disease diagnoses, user reviews, and other related content [3,4]. With the continuous development and prevalence of Internet-based healthcare, there is a growing demand among users for information in the healthcare domain. However, in online healthcare communities, obstacles such as information asymmetry [5,6] and trust issues [7] exist due to information overload and barriers to medical knowledge. In situations of information asymmetry, engaging in online transactions with strangers providing offline healthcare services poses risks for customers [6]. Therefore, online healthcare communities must strengthen their content development by delivering comprehensive, professional, and personalized content to users. This approach aims to enhance patient user stickiness and loyalty, attract more users, and increase their trust and satisfaction with the platform. The sustainable development of online medical consultation services stands as a paramount objective in this context [8].

Despite the increasing momentum in research on user-generated content in online healthcare, there has been limited attention to the sources of user-generated content and their connection to online patient decision-making [9,10]. Previous studies in the literature have explicitly examined the impact of user-generated content on patient decisions from aspects such as doctor characteristics [11,12], information quality [13], and patient perception [2,14]. Scholars have concluded that user-generated content originating from both doctors and patients is a major influencing factor in patient decision-making [4,15,16],

with evidence suggesting that the display of a doctor's professional background and online reputation in user-generated content plays a crucial role in improving online healthcare outcomes [17]. Despite this, there is still limited research in the literature on online healthcare that considers patient decision-making mechanisms from the perspective of user-generated content [3].

A notable characteristic of online healthcare is the virtual mode of information interactions, emphasizing the significance of user-generated content [3]. Purchasing medical consultation services in situations of information asymmetry may lead to adverse medical outcomes, such as personal safety and economic risks [5,6]. Therefore, trust is a crucial factor in overcoming uncertainty and reducing risks in online healthcare, with many patients choosing to establish trust in doctors through user-generated content on platforms [7,18]. However, there is a lack of in-depth research explaining how trust develops among different types of user-generated content [10]. In this context, our aim is to investigate the impact of various types of user-generated content on the purchase of online consultation services. In contrast to traditional healthcare, which emphasizes face-to-face communication, doctors in online healthcare communities leverage user-generated content to maintain competitiveness [19]. In online healthcare communities, the primary contributors to user-generated content include patient users and doctor users. Patients seek trust and support from both doctor-generated content and patient-generated content [10,15,19,20]. To systematically elucidate online patient behavior, there is a need for a contingency perspective to examine the boundary conditions of the user-generated content—the consultation service purchase relationship.

Indeed, price, responsiveness, and review consistency are three attractive factors influencing online patient decisions. In online healthcare, patients are sensitive to prices and may compare different medical services [21]. Price, as a crucial signal, signifies the economic cost risk of purchasing an unknown product [22]. Doctors with high-quality services and low prices may be more popular, as patients perceive greater value and trust in medical services [21,23]. On the other hand, responsiveness, as a signal of the doctor–patient interaction process, reflects the service efforts and proactiveness of doctors in remotely assisting patients with medical issues. In online healthcare communities, good responsiveness in interactions helps to build trust in doctors [24]. When doctors can promptly respond to patient inquiries, it enhances the quality of online medical consultations and patient satisfaction [25]. Responding to patient inquiries more promptly can reduce uncertainty in patients regarding medical advice and increase confidence in medical decision-making [26]. Moreover, consistency in reviews reflects the reliability of online reputation. Inconsistent reviews may increase the complexity of the shopping or decision-making process. Consumers may need more time to research and compare different reviews to make decisions, leading to user concerns or uncertainty about service quality [27,28]. For service providers, inconsistent reviews may create a negative impression of service reliability, thereby reducing trust [28,29]. While price, responsiveness, and review consistency are key trust-building signals of user-generated content, there is limited research in the literature studying their moderating effects on the interaction with user-generated content in influencing the purchase of online medical consultation services. To address this research gap, we examine the moderating effects of price, responsiveness, and review consistency on the user-generated content–consultation service purchase relationship from a trust perspective.

To achieve our research objectives, we employ the theory of trust to investigate the impact of user-generated content from different sources on patient decision-making. In online healthcare communities, user-generated content may encompass health knowledge, disease diagnoses, user reviews, and more. During the process of reviewing user-generated content, patients place trust in the opinions of doctors or peers, subsequently developing trust in doctors based on user-generated content from different sources. Additionally, we apply signal theory to examine the boundary conditions of the relationships between three types of user-generated content and the purchase of consultation services. According to signal theory, when information received by the signal recipient from the signal sender

is insufficient, information asymmetry occurs, leading to hesitation in decision-making by the signal recipient [30,31]. In such situations, patients (signal recipients) tend to seek external information cues to reduce the potential risks associated with medical services. In online medical consultations, price represents the cost of purchasing medical services and serves as an economic incentive influencing patients' choice of healthcare providers [32]. Responsiveness measures the extent to which doctors respond to patient inquiries, representing a signal of the quality of online interactions [33]. Review consistency reduces the risk of information overload and enhances the efficacy of evaluations [34–36]. Based on signal theory, we posit that signals, in conjunction with user-generated content, collectively reinforce patient trust in doctors and influence healthcare provider selection decisions.

We employ econometric methods to understand the impact of user-generated content on the purchase of consultation services and their interactions with various signals. Based on information from 10,000 doctor profiles across five cities, we indicate that user-generated content significantly influences patient selection behavior. Popular science articles and user ratings positively impact the purchase of consultation services, emphasizing the crucial role of doctors' knowledge-sharing and patient reviews in influencing customer decisions [10]. However, it is noteworthy that publicly displaying consultation records inhibits patient service purchases. This underscores the necessity for healthcare knowledge management [37] Furthermore, we find that pricing weakens the positive impact of popular science articles on the purchase of consultation services; doctor responsiveness mitigates the negative impact of publicly displaying consultation records on service purchases; and review consistency enhances the positive impact of patient ratings on the purchase of consultation services.

Our research contributes to several important areas. First, previous studies in the literature have emphasized the importance of trust in online healthcare communities [23,38]. However, trust based on user-generated content is crucial, as virtual communities fulfill patients' information needs. By examining the significance of user-generated content from different sources in the context of consultation service purchases, we reveal a trust mechanism based on user-generated content, consolidating and expanding trust theory. Previous research has partially examined the impact of user-generated content on consultation service purchases [7,11,15]. We take a holistic perspective to distinguish the roles of user-generated content from different sources, emphasizing that information dissemination through such platforms is pivotal for enhancing patient engagement and healthcare outcomes.

Second, this study contributes to signal theory by revealing the boundary conditions of the user-generated content—the consultation service purchase relationship. Previous research has used signal theory to explain user behavior in online healthcare communities, filling gaps in the existing literature regarding signal theory [20,39,40]. In this study, we provide empirical evidence answering the question of the moderating role of signals in determining consultation service purchases. Specifically, we find that patients are more likely to trust doctors with low prices, quick responsiveness, and consistent reviews.

Third, we add new knowledge to the literature on user-generated content in online healthcare communities. On the one hand, we reveal that user-generated content is a precursor to patient decision-making, considering different types of user-generated content together [10,11]. To our knowledge, this is the first investigation into the impact of different user-generated content on user purchase decisions in online healthcare communities. This not only supports the importance of user-generated content [17,41] but also confirms the prominent signals embedded in user-generated content [27,42]. Our findings further confirm a significant characteristic of online healthcare communities—the principle of doctors' information disclosures [43]

Fourth, the results of this study provide valuable recommendations for online patients, healthcare service providers, and platform policymakers to formulate service strategies and stimulate the vitality of online healthcare communities. Importantly, by understanding the dynamics of user-generated content and its impact on patient decision-making, stakeholders can develop initiatives that foster the sustainable growth of online healthcare

communities, ensuring their continued relevance and effectiveness in meeting the evolving needs of patients and healthcare providers.

## 2. Literature Review

### 2.1. User-Generated Content in Online Healthcare Community

Benefiting from the continuous improvement of user engagement on online healthcare community in recent years, numerous scholars have conducted extensive research on User-Generated Content (UGC) in the field of online healthcare [4,15,16,44]. User-Generated Content refers to information spontaneously created by users on online community portals [45]. In the context of online healthcare, sources of user-generated content include patient users and doctor users. User-generated content reflects preferences for healthcare services and reading this content can reduce information asymmetry between patients and healthcare service providers [46]. For example, Kordzadeh [4] argues that online reviews exhibit a high level of systematic bias, which may mislead potential patients and contradict the responsibility of healthcare service providers to act in the best interests of patients. User-generated content not only helps avoid low-quality service outcomes, increasing the likelihood of positive patient experiences, but also assists doctors in improving medical practices, as it directly reflects their service quality [47,48].

Previous research recognizes the positive impact of user-generated content on patient satisfaction and online healthcare service outcomes [3,15]. However, previous studies on online healthcare have mostly focused on patient-generated content such as online comments [4] and word-of-mouth [16,44]. Therefore, the research on different types of user-generated content and their differentiated effects remains relatively limited.

Some studies have delved into the factors influencing user-generated content from doctor users on online healthcare community [10–13]. These studies reveal that factors like a doctor's professional level, experience, qualifications, appointment transparency, service fees and response quality have moderating effects on doctor-generated content [11]. Through the exploration of doctor-generated content, the research has also discovered that information quality, emotional support, and source credibility significantly and positively impact patients' adoption of medical information [13]. Other antecedents stemming from doctor-generated content include the vocal features of doctor consultations [12] and knowledge-sharing [10].

Research on patient-generated content has primarily focused on online comments made by patients [20,49,50]. For instance, Chen and Baird [20] examined the impact of language signals in posts, including emotional valence, language style matching, readability, post length, and spelling, on the amount of social support patients receive. They found that emotional language signals, including negative emotions and language style matching, are effective in influencing patients in both obtaining information and emotional support from online communities. This indicates that the consensus of other patients on treatment experiences can influence a patient's perception of treatment outcomes [49], validating the idea that the perceived effectiveness of a treatment is closely related to the perception of community participants about the treatment. Additionally, certain types of language and other features of patients on medical portal websites may be associated with user behavior [50]. Prior work has also further discovered that the narrative authenticity and coherence of user texts can impact user decision-making behavior [51].

While these studies on user-generated content have directly explored the impact of individual user-generated content on patient perception and behavior, there are variations in focus on the user-generated content in online healthcare communities. However, while they all delve into hidden information within user-generated content to reveal user behavior, none has directly established an analytical model for the relationship between user-generated content and patient purchasing decisions. Therefore, further investigation into user-generated content in online healthcare communities is needed to uncover the differences in returns resulting from doctors' long-term efforts on their profile content [52].

## 2.2. Trust Theory

Trust theory posits that trust can be divided into two major categories: interpersonal trust and system trust, serving as an effective mechanism to reduce the complexity of social relationships [53]. Interpersonal trust refers to trust between individuals based on mutual acquaintance and emotional connection, while system trust is established on external mechanisms such as legal regulations and deterrents. In the online environment, consumer trust in service providers, specifically, their perception of a particular business's reputation, significantly influences consumers' willingness to make purchases [54]. Results from an online survey indicate that trust is the most critical factor in patients' selection of online healthcare services [55]. The broader goals of trust are to create positive impressions, ensure confidence in the reliability of the provider, and provide a sense of security during service usage or transactions [56].

Due to the increasing reliance of today's online consumers on UGC in making purchasing decisions, they tend to trust in, and rely more on, the UGC found in social media. Once customers establish trust in a seller or product through UGC, they are more likely to make related purchases of products or services [57]. For instance, on online platforms, potential customers are inclined to trust the opinions of previous customers who express their experiences through reviews when making purchasing decisions [7]. Simultaneously, the knowledge shared by doctors also influences patients' behavioral decisions [10,11].

Trust is a critical factor in online healthcare communities [2,18] due to the apparent issue of information asymmetry between online healthcare service providers and customers [5,6]. User-generated content plays a crucial role in building trust, as it provides potential customers with real-life experiences from authentic users, enabling others to better understand and evaluate specific products or services [58]. Scholars have identified that user-generated content from both doctors and patients is a key dimension influencing overall patient satisfaction [10,15,20]. Therefore, we apply trust theory to elucidate how different types of user-generated content contribute to trust formation in the context of consultation service purchases.

## 2.3. Signaling Theory

Signal theory analyzes how individuals with information advantages in the market can transmit information credibly to those in information disadvantages through "signal transmission" to address the problem of achieving efficient market equilibrium under conditions of information asymmetry [31].

Based on a review of the literature on the application of signal theory, the theory comprises three elements: signal sender, signal, and receiver [59]. In situations of information asymmetry, signal senders can choose to reduce information asymmetry by sending signals to receivers. Signal quality is defined by the cost of the signal (the cost of preparing the signal) and its value to the receiver. Alternatively, receivers can choose to reduce information asymmetry by searching for additional information. The decision to search depends on the cost and value of the information. Signal theory posits that the decision-making behavior of signal receivers is often dominated by the signals received. Natural information asymmetry between signal receivers and senders leads to issues, such as adverse selection, moral hazard, and credit risk, due to the inherent asymmetry of information. Parties with information advantages and disadvantages attempt to use signals to convey "true" information they hold to the other party [31,59].

Signal theory, applied in various contexts, explains the impact of information asymmetry in different backgrounds [42,60–62]. Online healthcare, being a unique electronic commerce community centered around medical services as products, also grapples with information asymmetry. Physicians meticulously manage and maintain information on their personal profiles, such as clinical titles, to send reliable signals to potential patients, aiming to gain more economic returns and social reputation. Scholars have utilized signal theory to investigate the impact of user-generated content on patient behavior in various online healthcare consultation service scenarios. For instance, Ref. [9] applied signal theory,

indicating that two signals (doctor's professional status and service feedback), have a significantly positive influence on patients' choice of doctors. This underscores the importance of doctors, as signal senders, conveying signals to signal receivers about their excellent professional status. Similarly, ref. [39] employed signal theory to demonstrate that reputation diversity and experience diversity of doctors in online health community teams have a positive impact on team performance.

Other signals also include doctors' favorability ratings [41], the fulfillment of demands [24], pricing, and responsiveness [42]. Pricing, as an effective signal for medical services, not only reflects service quality but also indicates the cost of purchasing services [21,63]. Furthermore, the doctor–patient interaction process is a crucial avenue for patients to form initial impressions of physicians [40], and responsiveness serves as an appealing signal, illustrating the quality of the interactions in online medical communication [42]. Consistency in reviews reflects the reliability of user evaluations [27]. Therefore, building on the signal theory, we will consider the moderating effects of signals such as pricing, responsiveness, and review consistency. The theoretical model for the study is illustrated in Figure 1.

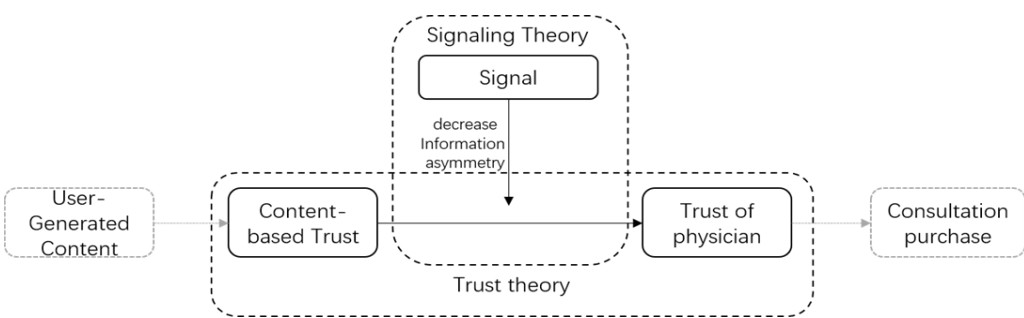

**Figure 1.** Theoretical research model.

## 3. Hypotheses

### 3.1. The Impact of User-Generated Content on Consultation Purchase

Doctors, as healthcare service providers, are crucial sources of information for patients when choosing healthcare providers [43]. Sharing medical knowledge contributes to the gaining of better social reputations and to greater economic rewards for doctors [27]. Doctors proactively share popular science articles, introducing and popularizing health knowledge, scientific research, and medical advancements. Studies have found that the knowledge shared by doctors also influences patients' decision-making [10,11]. In today's consumer landscape, when making purchasing decisions, consumers increasingly trust and rely on the user-generated content in social media, and they are more likely to purchase services they trust [57].

In the online healthcare environment, patients not only need to understand their specific illnesses and treatment options but also seek reliable medical resources to enhance their treatment and recovery [52]. Existing research underscores that trust is a pivotal factor in online healthcare communities, given the apparent issue of information asymmetry between online healthcare service providers and customers [5,6]. Patients benefit from reading professionally shared information by doctors, aiding their understanding of the doctors' medical proficiency and fostering trust. Popular science articles grounded in disease knowledge and treatment experience serve as reflections of the doctors' expertise in their respective fields [17].

Therefore, we hypothesize that doctors publishing professional articles on their profiles leads patients to believe that the doctors possess substantial knowledge and experience. This, in turn, enhances patients' trust in the doctors, leading them to believe that they will receive better medical services.

**Hypothesis 1 .** *Doctors who publish more science articles will receive more consultation purchase.*

Many patients, when seeking medical services, often browse through the displayed past treatment dialogues on a doctor's profile to better understand the doctor's professional background and treatment experience [64]. Online platform users tend to trust the opinions of others [65]. These dialogues provide potential clients with real user experiences, helping patients to establish trust with doctors and significantly influencing patients' medical decisions [66].

Individual choices are influenced by the experiences of others [67]. However, the extent to which doctors openly displaying past consultation dialogues contributes to trust-building is not clear. Contributors to consultation dialogues include both doctors and past patients, and the conversion of content-based trust into trust in doctors might be challenging [68]. On the contrary, disease-related diagnostic and treatment suggestions in the content may lead patients to plan their own treatment, thereby reducing consultation purchases. Therefore, we hypothesize that:

**Hypothesis 2.** *Doctors who display more consultation dialogues will receive fewer consultation purchases.*

Patient reviews serve as an information-filtering mechanism, helping patients to find suitable healthcare service providers among numerous doctors [41]. Online reviews are critical to building trust, as they convey clues to potential patients about the quality of a physician's care. Scholars have found that by extracting useful clues from online reviews, patients can reduce perceived risk and increase trust in their physicians when choosing a counseling service [14].

For patients, doctors with high ratings indicate good reputations and professional competence in the medical field. Once patients develop trust in doctors from online reviews, they are likely to purchase related products or services. Evidence suggests that on online platforms, prospective customers tend to trust previous users who share their experiences in online reviews [7]. Perceived healthcare service quality from online reviews assures people about their health needs and fosters trust in doctors [69] Additionally, trust in doctors promotes physician selection, with overall ratings and the volume of reviews significantly positively correlated with the willingness to choose a physician [70,71]. In other words, higher patient ratings will facilitate the construction of trust in doctors and promote the purchase of consultation services. Therefore, we hypothesize that:

**Hypothesis 3.** *Doctors with higher patient ratings will receive more purchases of consultation services.*

### 3.2. Interaction Effect of Signals

#### 3.2.1. Signal of Price

Price has a remarkable signaling effect on the presence of information asymmetry [72,73]. Due to information asymmetry in online transactions, consumers may seek multiple cues to help them make a purchase decision, with low price being one of the most characteristic attractions [74].

The quality of healthcare services significantly influences patients' purchases of consultation services [75]. Price represents the economic cost that patients pay for purchasing healthcare services [76]. If a doctor's prices are too high and exceed what patients can afford, patients may choose other doctors with more reasonable prices, even if their ratings are lower [21] That is, price implies monetary sacrifice and financial risk in purchasing an unknown service, and, in order to minimize the potential financial risk, consumers usually compare the prices of online services over and over again and choose a reasonable price.

User willingness–behavior relationships in online communities are often influenced by price [21,77]. Research has demonstrated an interactive effect between price signals and healthcare service quality [21,63] Popular science articles are part of healthcare services, and patients evaluate both the quality of healthcare services and the required prices during

the process of receiving the knowledge shared by doctors. Low prices indicate cost savings and act as an economic incentive, influencing patients' choices of healthcare providers [32].

Based on signaling theory, we argue that price can weaken the positive effect of professional articles on the purchase of consulting services [23]. If the price of healthcare services is low, consumers may perceive greater benefits (the difference between perceived value and cost) and be more willing to choose healthcare services [78]. Additionally, a lower price indicates less financial risk associated with purchasing the service, creating greater economic incentives, and patients are generally more inclined to choose lower-priced healthcare services [79]. Secondly, service prices are often considered to represent service quality, and lower-priced services are not typically associated with high service quality [80]. Once lower-priced services involve more knowledge sharing, patients may feel more satisfied because the service exceeds expectations. Therefore, we hypothesize that:

**Hypothesis 4.** *The relationship between science articles and the consultation purchase is moderated by price. This relationship is expected to strengthen at a lower price level.*

3.2.2. Signal of Responsiveness

Another appealing factor in online healthcare is the responsiveness of doctor–patient interactions, a crucial service signal influencing user decisions [24]. Responsiveness describes the quality of the interactions between patients and doctors, focusing on the speed with which doctors respond to patient inquiries [33].

Many patients, when choosing healthcare service providers, browse through doctors' consultation records. These records encompass interactive information, including the text, images, and audio exchanged between physicians and other patients concerning medical conditions. Response time emerges as a pivotal determinant in online doctor–patient interactions. Existing research emphasizes that doctor–patient interactions are a significant factor influencing patients' choice of healthcare providers [40] Doctor responsiveness, a notable variable in the doctor–patient interaction process, indicates a willingness and ability to address patients' health concerns and provide high-quality service. High levels of responsiveness also suggest doctors actively engage in online interactions, offering necessary care and advice, thereby enhancing patient satisfaction during the medical consultation process. Additionally, timely responses reduce uncertainty about medical advice, boosting patient confidence in medical decisions [26].

Based on signaling theory, we argue that responsiveness can weaken the negative impact of consultation records on the purchase of consulting services. First, in the field of online medical consultations, responsiveness can convey that the provider cares, which is an important factor in building physician trust [24]. Second, responsiveness in consulting services may represent the doctor's capability. When doctors can promptly respond to patients' inquiry needs, it enhances the quality and patient satisfaction of online medical consultations [25]. Third, from the perspective of social interactions, outstanding responsiveness in consultation dialogues implies smooth online communication, thereby improving the perceived reliability of patients and reducing the perceived risk [81]. Responsiveness vividly demonstrates the interaction quality of healthcare service providers both offline and online, vividly describing the professional attributes of doctors, and further enhancing trust in healthcare service providers [38]. When patients browse through past consultation dialogues, highly responsive dialogues contribute to building trust in doctors. Therefore, we hypothesize that:

**Hypothesis 5.** *The relationship between publicly displaying consultation records and the consultation purchase is moderated by responsiveness, and this relationship will be weakened at a higher level of responsiveness.*

### 3.2.3. Signal of Consistency

Patients are more likely to choose highly rated physicians for their consultations because patient ratings convey information about the quality of the medical service [41]. However, patient evaluations are highly skewed and exhibit systematic biases that may mislead patients [4].

Relying on online reviews for purchasing decisions has become a common practice for modern consumers [57]. Online reviews provide insights into the actual user experiences of products or services, which is helpful in deciding whether to make a purchase. Additionally, third-party opinions contribute to users' trust in service providers [7] Studies confirm that inconsistent reviews may lead to user concerns about the quality of services [27,28]. Inconsistent reviews may increase the complexity of the shopping or decision-making process. Consumers need to spend more time comparing different reviews to make decisions. On the other hand, for service providers, inconsistent reviews may create a negative impression of service reliability, reducing trust [28,29].

Based on signaling theory, we posit that the consistency of reviews can enhance the positive effect of patient ratings on the purchase of consulting services. First, reviews are crucial for maintaining trust, reducing risk, and providing a better user experience and service [15,16]. User reviews offer feedback on the quality of healthcare service providers, and when patients see positive evaluations and shared experiences from others, they are more likely to trust the doctor's professional knowledge and abilities. Second, consistent reviews convey consensus and consistent viewpoints, reducing the amount of information consumers need to process and thereby lowering the risk of information overload [34,35].Third, further evidence suggests that the impact of review efficacy is highly dependent on its consistency with other available review efficacies [36]. Therefore, we hypothesize that:

**Hypothesis 6.** *The relationship between patient ratings and the consultation purchase is moderated by review consistency, where this relationship is strengthened at a higher level of consistency.*

To sum up, we formulate research hypotheses that affect the selection behavior of patients in online health communities, as shown in Figure 2.

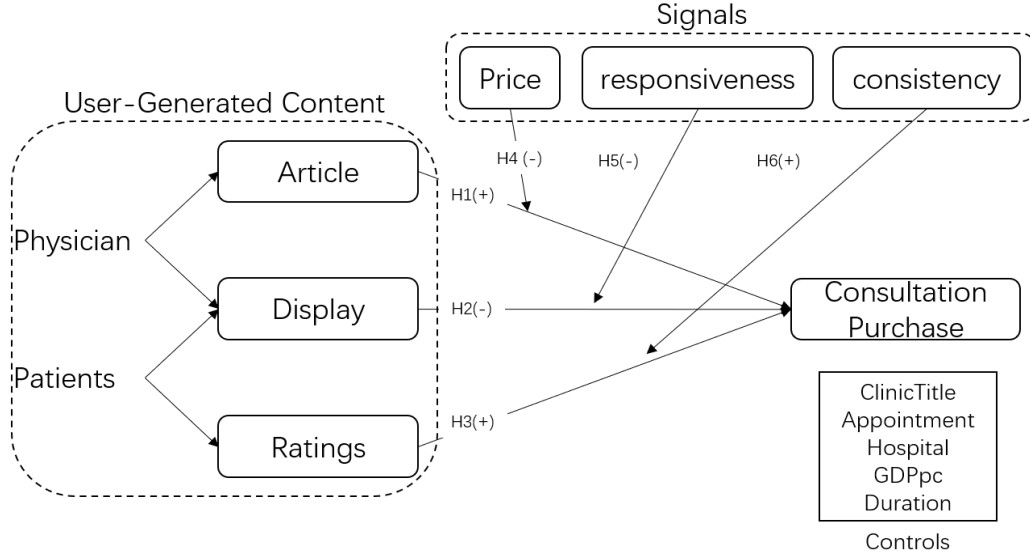

**Figure 2.** Research framework.

## 4. Research Methodology

### 4.1. Data Resources

Haodf.com is the largest medical website in China, covering nearly 500,000 doctors from over 6000 hospitals. It serves as a platform for Chinese patients to obtain medical

consultations remotely from doctors, allowing them to provide comments or opinions on doctors from different regions. The consultation fees are determined independently by the doctors, with many adopting a fee-based model for their services. Typically, more renowned doctors charge higher fees for individual consultations.

Selecting Haodf.com for this study is justified for three reasons: (1) representativeness: Haodf.com is the most widely used online healthcare platform in China, with millions of patients having received services on the platform, including online text consultations, phone consultations, outpatient appointments, and expert team consultations. Studying this platform is representative of the broader landscape; (2) professionalism: Haodf.com is a specialized online healthcare service platform with over 610,000 doctors from more than 9900 different hospitals. Many of these doctors are from the highest professional-level hospitals in China; and (3) alignment with research objectives: Haodf.com openly displays authentic user-generated content and user behavioral records, providing reliable data support for our research.

The economic development levels and the list of top-tier hospitals in each region are sourced from official Chinese government websites.

### 4.2. Sample and Data Collection

Data were collected via a Python program between October and December 2022. Initially, we identified the profiles of 10,000 doctors from five representative regions, Beijing, Shanghai, Shandong, Henan, Hubei, each with 2000 doctors. Subsequently, we gathered personal information, hospital details, user-generated content, and patient selection information from the doctors' profiles. Finally, we matched the hospitals and local economic development levels based on the list of top-tier hospitals publicly available on the Chinese government's website and the per capita GDP in each region. This process resulted in a cross-sectional dataset comprising 10,000 samples.

#### 4.2.1. Dependent Variable

We designate patient consultation (Consultation) as the dependent variable, indicating which doctor a patient opts for in the online health community when seeking medical inquiries. On Haodf.com, each doctor's individual profile exhibits the total count of online consultation services provided. Hence, we employ the doctor's online consultation volume as a measure with which to assess patient preferences for a particular doctor.

#### 4.2.2. Independent Variable

We consider three types of user-generated content: popular science articles (doctor-generated content), publicly displayed consultation records (interaction-generated content), and patient ratings (patient-generated content). These types of content reflect a doctor's long-term efforts, and patients can autonomously review them on the doctor's profile before making an inquiry decision. We utilize the number of popular science articles (Article) published by a doctor as the independent variable for doctor-generated content. As some doctors' consultation records can be publicly accessed, there is collinearity between the number of consultation records and the quantity of online consultation services, making this unsuitable for inclusion in the model. Therefore, we constructed the independent variable "Display" to assess the behavior of doctors publicly displaying consultation records. If the proportion of publicly displayed consultation records by a doctor exceeds the mean value, it is assigned a value of 1; otherwise, it is assigned 0. Another patient-generated content independent variable is patient ratings (Recommendation), representing the average satisfaction with efficacy and attitude, as rated by patients.

#### 4.2.3. Moderating Variables

The doctor's profile displays the consultation service prices, and typically, renowned doctors from major hospitals have higher prices. The frequently observed price signal reflects economic costs and its interaction with other factors. Responsiveness signal indicates

the doctor's response speed, while consistency signal in reviews reflects the reliability of peer comments. Therefore, we consider consultation price (Price), responsiveness (Responsiveness), and review consistency (Consistency) as moderating variables. We define review consistency as:

$$Consistency_i = log\left(\frac{1}{|Efficacy_i - Attitude_i|}\right) \tag{1}$$

Efficacy and Service represent patient satisfaction with the effectiveness and attitude of the doctor, respectively.

### 4.2.4. Controls

The model also includes other variables as control factors that influence patient selection behavior: the hospital level where the doctor works (Hospital), the clinical title of the doctor (Clinic), whether the doctor provides offline registration services on the platform (Offline), regional economic development level (GDPpc), platform registration duration (Duration), and so on. These variables are used to control the influencing factors of patient selection behavior in the research model. Table 1 provides descriptions of all variables.

**Table 1.** Descriptive statistics analysis of variables.

| Variable | Mean | Std. Dev. | Min | Max | Definition |
|---|---|---|---|---|---|
| Consultation | 2452.2 | 4328.9 | 0 | 64,998 | The quantity of consultation orders received by doctors |
| Gift | 166.5 | 408.5 | 0 | 10,209 | The quantity of gifts received by doctors |
| Article | 34.1 | 135.9 | 0 | 6153 | The number of scientific articles published by doctors. |
| Display | 0.4 | 0.5 | 0 | 1 | The proportion of doctors who publicly display their consultation records. Greater than the average = 1, else = 0 |
| Recommendation | 3.9 | 0.4 | 3.3 | 5 | The overall recommendation score by the system. |
| Price | 61.6 | 99.3 | 0 | 3000 | The price of one medical consultation. |
| Responsiveness | 4.0 | 1.4 | 1 | 5 | The speed of doctors' responses. |
| Consistency | 3.3 | 2.0 | 0 | 4.61 | The consistency between satisfaction with the effectiveness of the doctor's treatment and satisfaction with the doctor's attitude. |
| ClinicTitle | 3.4 | 0.7 | 1 | 4 | Chief Physician = 4, Associate Chief Physician = 3, Attending Physician = 2, Other = 1 |
| Appointment | 0.5 | 0.5 | 0 | 1 | Enabled online appointment service = 1, else = 0 |
| Hospital | 1.0 | 0.2 | 0 | 1 | Tertiary hospital = 1, Non-tertiary hospital = 0 |
| GDPpc | 12.2 | 5.2 | 6.2 | 19.0 | Per capita GDP |
| Duration | 3292.9 | 1391.2 | 12 | 5435 | Number of days since the doctor opened an account on the platform |

The online consultation volume and the quantity of gifts exhibit large magnitudes and skewed distributions. The clinical title level of doctors involves categorical data, and these three variables may demonstrate a nonlinear impact on patient selection behavior. Thus, the natural logarithms of these three variables are taken and incorporated into the model.

Table 2 presents the correlation coefficients of the variables. As shown in Table 2, all independent variables exhibit significant correlations with the dependent variable, aligning with our hypotheses. The correlation coefficients among variables are all below 0.8, within a reasonable range, and are not expected to impact the model estimation results [82].

**Table 2.** Correlation matrix of variables.

| Variable | (1) | (2) | (3) | (4) | (5) | (6) | (7) | (8) |
|---|---|---|---|---|---|---|---|---|
| Consultation | 1 | | | | | | | |
| Gift | 0.744 *** | 1 | | | | | | |
| Article | 0.332 *** | 0.275 *** | 1 | | | | | |
| Display | −0.117 *** | −0.068 *** | −0.035 *** | 1 | | | | |
| Recommendation | 0.465 *** | 0.388 *** | 0.202 *** | 0.049 *** | 1 | | | |

**Table 2.** *Cont.*

| Variable | (1) | (2) | (3) | (4) | (5) | (6) | (7) | (8) |
|---|---|---|---|---|---|---|---|---|
| Price | 0.254 *** | 0.272 *** | 0.077 *** | 0.062 *** | 0.262 *** | 1 | | |
| Responsiveness | 0.027 ** | 0.007 | 0.014 | −0.012 | 0.058 *** | 0.027 ** | 1 | |
| Consistency | 0.030 *** | 0.067 *** | −0.004 | 0.031 *** | −0.200 *** | 0.023 ** | 0.144 *** | 1 |

(1) = Consultation, (2) = Gift, (3) = Article, (4) = Display, (5) = Recommendation, (6) = Price, (7) = Responsiveness, (8) = Consistency. *** $p < 0.01$, ** $p < 0.05$.

### 4.3. Estimation

To test the research hypotheses, we employed ordinary least squares (OLS) regression and established the following multivariate regression model:

$$
\begin{aligned}
Income_i&[log(Consultation_i)] \\
&= \alpha_0 + \alpha_1 Article_i + \alpha_2 Display_i + \alpha_3 Recommendation_i + \alpha_4 Article_i \times Price_i \\
&+ \alpha_5 Display_i \times Responsiveness_i + \alpha_6 Recommendation_i \times Consistency_i + \alpha_7 Price_i \\
&+ \alpha_8 Responsiveness_i + \alpha_9 Consistency_i + \alpha_{10} GDPpc_i + \alpha_{11} log(ClinicTitle_i) \\
&+ \alpha_{12} Appointment_i + \alpha_{13} Hospital_i + \alpha_{14} Duration_i + u_i
\end{aligned}
\tag{2}
$$

Here, *Article*, *Display*, *Recommendation* are independent variables, *Price*, *Responsiveness*, *Consistency* are moderating variables, Random error is indicated by $u_i$.

### 4.4. Empirical Results

#### 4.4.1. Main Effects

We conducted multiple regression analysis using STATA, and Table 3 provides the standardized estimation of consultation purchases. All models exhibit adjusted R-squared values greater than 0.30, and the F-statistics (exceeding 150) are both reasonable and sufficiently large. To test for multicollinearity, we performed variance inflation factor (VIF) tests. The VIF values for each independent variable are all less than 10, indicating the absence of multicollinearity issues, rendering the models reasonable and significant [83].

**Table 3.** Standardized estimation results.

| Log(Consultation) | Model (1) | Model (2) | Model (3) | Model (4) | Model (5) |
|---|---|---|---|---|---|
| Article | 0.001 *** (0.000) | 0.002 *** (0.000) | 0.001 *** (0.000) | 0.001 *** (0.000) | 0.001 *** (0.000) |
| Display | −0.595 *** (0.037) | −1.454 *** (0.036) | −0.919 *** (0.098) | −1.38 *** (0.035) | −0.908 *** (0.098) |
| Recommendation | 1.223 *** (0.039) | 1.524 *** (0.044) | 1.285 *** (0.038) | 0.522 *** (0.065) | 0.658 *** (0.062) |
| Article * Price | | −0.000 *** (0.000) | | | −0.000 ** (0.000) |
| Display * Responsiveness | | | 0.088 *** (0.023) | | 0.084 *** (0.023) |
| Recommendation * Consistency | | | | 0.352 *** (0.018) | 0.180 *** (0.018) |
| Price | 0.002 *** (0.000) | 0.003 *** (0.001) | | | 0.002 *** (0.000) |
| Responsiveness | 0.023 ** (0.011) | | 0.001 (0.012) | | −0.016 (0.012) |
| Consistency | 0.040 *** (0.007) | | | −1.460 *** (0.071) | −0.685 *** (0.072) |
| ClinicTitle | −0.059 ** (0.023) | −0.106 *** (0.026) | −0.008 (0.022) | −0.039 (0.025) | −0.055 ** (0.023) |
| Appointment | 0.175 *** (0.035) | 0.481 *** (0.038) | 0.146 *** (0.034) | 0.456 *** (0.037) | 0.168 *** (0.035) |
| Hospital | −0.440 *** (0.071) | −0.541 *** (0.086) | −0.383 *** (0.072) | −0.496 *** (0.085) | −0.427 *** (0.070) |
| GDPpc | 0.027 *** (0.004) | 0.088 *** (0.005) | 0.035 *** (0.004) | 0.087 *** (0.004) | 0.026 *** (0.004) |
| Duration | 0.000 *** (0.000) | 0.000 *** (0.000) | 0.000 *** (0.000) | 0.000 *** (0.000) | 0.000 *** (0.000) |
| Constant | 1.057 *** | −0.740 *** | 0.725 *** | 3.215 *** | 3.443 *** |
| Observations | 5441 | 7382 | 5523 | 8026 | 5441 |
| $R^2$_adjusted | 0.449 | 0.558 | 0.435 | 0.564 | 0.462 |
| Max VIF | 1.61 | 2.87 | 9.77 | 8.82 | 9.91 |
| F test | 373.1 *** | 766.0 | 366.5 | 894.1 | 302.6 |

Robust t-statistics in parentheses. The fluctuation in data volume stems from missing variable data. *** = $p < 0.01$, ** = $p < 0.05$, * = $p < 0.1$.

Model 1 incorporates control variables, independent variables, and moderating variables. Model 2 considers the interaction between price and popular science articles. Model 3 examines the interaction between responsiveness and the public display of inquiry records. Model 4 explores the interaction between consistency in reviews and patient ratings. Model 5 represents the comprehensive model, incorporating simultaneous adjustments for all price, responsiveness, and consistency factors. To address multicollinearity concerns, we employed a logarithmic transformation on the 'consistency'.

These five models consistently demonstrate that the valence of Article ($p < 0.01$) and that of Ratings ($p < 0.01$) have a positive and significant effect on Consultation, supporting Hypotheses 1 and 3. These findings confirm that the knowledge sharing (Article) and user recommendation (Ratings) are crucial in building patient trust and determining their purchases [14,17]. However, the public display of inquiry records ($p < 0.01$) is associated with a decrease in patients' consultation service purchases. Thus, Hypothesis 2 is supported.

4.4.2. Estimation of Interaction Effects

The full model indicates that price negatively moderates the positive effect of knowledge sharing on consultation service purchases (beta = $-0.000$, SE = 0.000, $p < 0.05$). To illustrate the interaction effect of price more clearly, Figure 3 depicts the interaction of price and articles on consultation purchases. Price is divided into high and low categories based on the average value plus or minus one standard deviation. The results demonstrate that price weakens the positive impact of popular science articles on consultation service purchases, supporting Hypothesis 4. The findings suggest that low prices align with a higher level of knowledge sharing, which better meets the needs of patients, consistent with previous research conclusions [27,63].

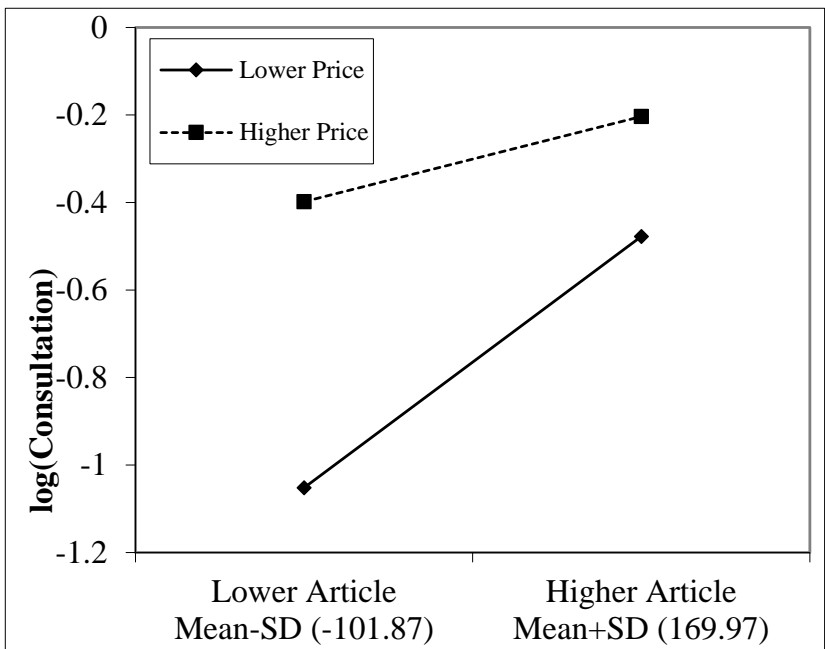

**Figure 3.** Interaction effect of price and articles.

Figure 4 presents the interaction effect of responsiveness and display on consultation. We divided responsiveness into high and low categories based on one standard deviation above and below the mean. The results indicate that doctor responsiveness weakens the negative impact of displaying consultation records on consultation service purchases. Responsiveness signals contribute to building trust in doctors [24]. Thus Hypothesis 5 is supported.

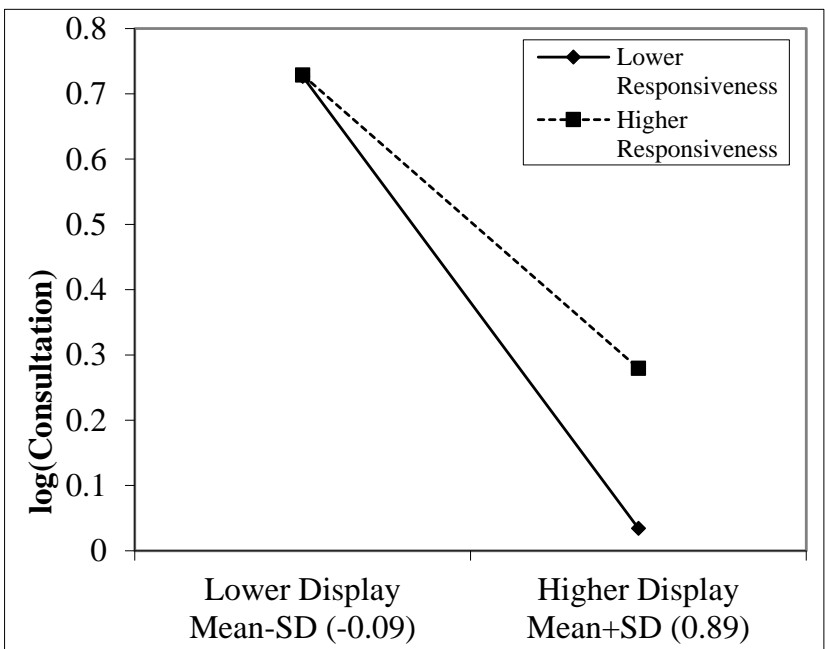

**Figure 4.** Interaction effect of responsiveness and display.

Figure 5 presents the interaction effect of consistency and ratings. We divided comment consistency into high and low categories based on one standard deviation above and below the mean. The results indicate that comment consistency positively moderates the positive impact of patient ratings on consultation service purchases. Specifically, the positive effect of patient ratings on consultation service purchases is greater for doctors with high comment consistency compared to those with low comment consistency. In other words, comment consistency signals enhance the reliability of comments, thereby strengthening trust [36]. Thus Hypothesis 6 is supported.

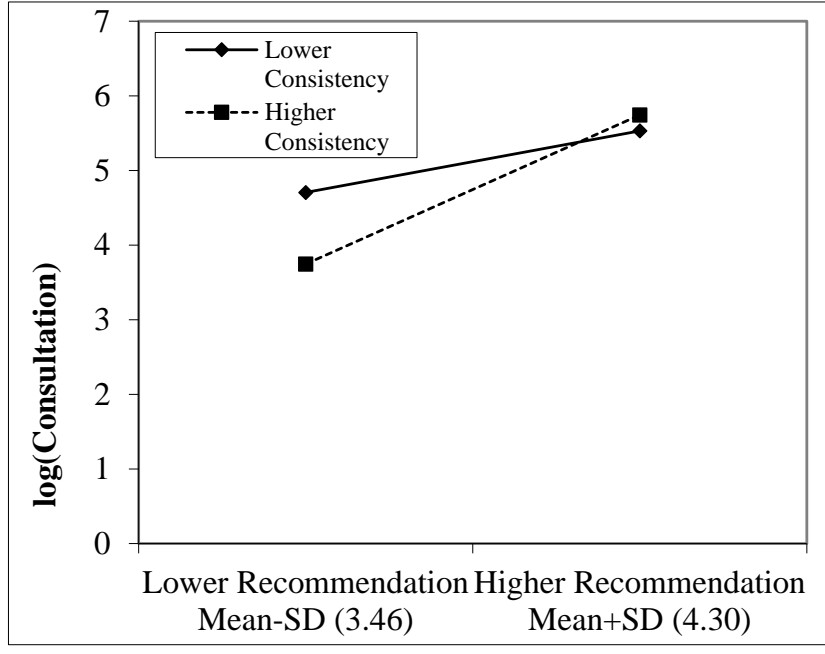

**Figure 5.** Interaction effect of consistency and ratings.

### 4.5. Robustness Checks and Additional Analyses

We conducted multiple tests to ensure the robustness of the proposed model, as shown in Table 4. First, following previous studies in the literature, a gift can be regarded as the proxy of economic expenditure [84]. we replace Consultation with Gift to measure payment willing [85], as shown in Model 1. Second, due to the dispersed distribution of Price, we transform Price from a continuous variable to a categorical one (1 = free, 2 = large than 0 USD and less than 1.38 USD, 3 = large than 1.38 USD and less than 4.14 USD, 4 = large than 4.14 USD and less than 8.29 USD, 5 = large than 8.29 USD), as illustrated in Model 2. Third, we take a negative binomial regression to repeat our proposed model because Consultation is non-negative count data, as presented in Model 3. Multicollinearity was detected in the model due to the inclusion of interaction terms. Following a logarithmic transformation of the variables, the maximum VIF for all models remained below the threshold of 10. Overall, these results are consistent with our findings in the original model.

**Table 4.** Robust check.

| | Model (1): Replacing Consultation Volume with Gift Volume | Model (2): Changing "Price" from a Continuous Variable to a Categorical One | Model 3: Using a Negative Binomial Regression |
|---|---|---|---|
| Article | 0.001 *** (0.000) | 0.228 *** (0.009) | 0.032 *** (0.001) |
| Display | −0.806 *** (0.110) | −0.764 *** (0.091) | −0.112 *** (0.013) |
| Recommendation | 0.521 *** (0.074) | 0.917 *** (0.038) | 0.12 *** (0.005) |
| Article * Price | −0.000 (0.000) | −0.000 (0.000) | −0.000 ** (0.000) |
| Display * Responsiveness | 0.056 ** (0.026) | 0.061 ** (0.021) | 0.009 ** (0.003) |
| Recommendation * Consistency | 0.244 *** (0.020) | 0.018 *** (0.002) | 0.002 *** (0.000) |
| Price | 0.002 *** (0.000) | 0.243 *** (0.016) | 0.034 *** (0.002) |
| Responsiveness | −0.031 ** (0.015) | −0.015 (0.032) | −0.002 (0.004) |
| Consistency | −0.896 *** (0.082) | −0.089 *** (0.013) | −0.013 *** (0.002) |
| ClinicTitle | −0.015 (0.026) | −0.092 *** (0.022) | −0.013 *** (0.003) |
| Appointment | 0.054 (0.040) | 0.061 (0.032) | 0.010 * (0.005) |
| Hospital | −0.120 (0.091) | −0.36 *** (0.067) | −0.048 *** (0.009) |
| GDPpc | 0.061 *** (0.004) | 0.012 ** (0.004) | 0.002 *** (0.001) |
| Duration | 0.000 *** (0.000) | 0.000 *** (0.000) | 0.000 *** (0.000) |
| Constant | −0.094 | 1.669 *** | 1.217 *** |
| Observations | 5352 | 5523 | 5523 |
| $R^2$_adjusted | 0.471 | 0.518 | - |
| Wald chi$^2$ | - | - | 5054.58 |
| Pseudo $R^2$ | - | - | 0.04 |
| Max VIF | 9.74 | 9.54 | 9.54 |
| F test | 346.7 *** | 318.1 *** | - |

Standardized BRs are reported with robust standard errors in parentheses. *** = $p < 0.01$, ** = $p < 0.05$, * = $p < 0.1$.

## 5. Findings and Discussion

### 5.1. Key Findings

Drawing from signal theory and trust theory, we address the following two key questions regarding user-generated content: (1) How does user-generated content influence patients' decisions to purchase consultation services?; and (2) What moderating effects do external signals have on different types of user-generated content? We construct multiple regression analysis models controlling for fixed effects such as appointment channels, physician titles, and regional development levels to empirically test our hypotheses. Utilizing data collected from "Haodf.com", we uncover the effects of various user-generated content types:

1.  Doctors who publish more popular science articles will receive more purchases of consultation services;
2.  Doctors who display fewer consultation records tend to receive fewer purchases of consultation services;

3. Doctors with higher patient ratings tend to receive more purchases of consultation services.

The findings confirm our theoretical model. When doctors publish popular science articles on their homepage, it signals to patients that they possess considerable knowledge and experience, as well as the ability to provide high-quality medical services. Patients respond to this signal by trusting that they will receive better medical care, thereby increasing their likelihood of purchasing services. Similarly, positive patient testimonials have a similar effect. This underscores the importance of doctor knowledge-sharing and patient ratings in influencing customer decisions [10].

Doctors publicly displaying their past consultation records can indeed showcase their medical skills, enhancing patients' understanding and trust in them [11]. However, on the other hand, consultation records represent information generated from interactions between doctors and past patients. Patients can directly access specific medical knowledge about their own conditions from these records, enabling them to plan their treatment independently. This diminishes their inclination to engage in further consultations with the doctor, resulting in reduced purchases of consultation services. This finding does not negate the importance of doctor–patient interactions [24,81], but rather emphasizes the need to protect intellectual property rights in the doctor–patient interaction process.

Additionally, the experimental results elucidate the interactive effects of external signals:

4. Price attenuates the positive impact of knowledge sharing (Article) on consultation service purchases;
5. Responsiveness mitigates the negative impact of publicly displaying consultation records on consultation service purchases;
6. Consistency strengthens the positive impact of patient ratings on consultation service purchases.

This indicates that additional signals from various forms of user-generated content can influence customer trust and impact sales performance [32,72].

The experiment reveals that the price, perceived by patients as a signal of high service quality [73], aligns with the signal of knowledge-sharing articles. In other words, medical consultation prices reinforce patients' expectations of medical service outcomes.

### 5.2. Theoretical Implications

The rapid development of online healthcare reflects the continuously growing demand for information in the medical and health fields among users [3,15]. To gain a deeper understanding of user-generated content in online healthcare communities, we identified three key dimensions of user-generated content (i.e., knowledge-sharing articles, consultation records, user reviews) and studied the effects of different types of user-generated content on service purchasing from a trust perspective. Our study contributes to the theoretical literature in several ways. First, by elucidating the importance of trust in various user-generated content in online healthcare, we provide a nuanced insight into trust theory. We advance this understanding by demonstrating, for the first time, how trust in healthcare service providers is independently driven by different types of user-generated content, thereby advancing this knowledge. Previous studies have utilized trust theory to elucidate trust in doctors from various perspectives, including doctor characteristics [11], patient characteristics [64], and overall satisfaction [10,20,23]. Furthermore, trust based on user-generated content is crucial, as virtual communities meet users' information needs. However, in online healthcare, little attention has been paid to trust based on different types of user-generated content [7,11,15]. We emphasize that the information generated by both doctors and patients in the past can help potential users build trust in doctors, thus influencing patients' payment decisions.

Second, we contribute to signal theory by revealing the boundary conditions of the relationship between user-generated content and consulting service purchases. Previous research has applied signal theory to explain user behavior in online medical communities [20,39,40]. The viewpoint of interpreting user choice from the perspective of signal

theory complements existing research. For example, some scholars suggest that signals of doctor attributes may attract customers' attention and influence patient decisions [40]. Some studies suggest that various reliable signals can complement each other, enhancing customer trust [39,40]. In this study, we provide empirical evidence to answer how signals act on user-generated content to influence user decisions. Specifically, we have demonstrated that signals from various sources (such as doctor's articles and patient reviews) can affect patient trust and promote consulting service purchases. From the perspective of services, we found that patients are more likely to trust doctors with lower prices, quick responsiveness, and consistent ratings. This elucidates the importance of trust as a factor in building patient trust, as previously proposed in research [23,38]. Overall, we have filled a gap in the existing literature regarding signal theory, revealing how different signals can collectively influence customer trust, and, consequently, impact consulting service purchases.

Third, this research contributes new insights to the literature on user-generated content in online healthcare communities. On the one hand, we reveal user-generated content as a precursor to patient decision-making. Previous research on online healthcare has examined the impact of user reviews on patient choice of healthcare providers [4,16,44]. These studies emphasize patient reviews as significant indicators influencing patient purchase behavior. However, there has been limited research on the impact of physician-generated content on patient purchase behavior, necessitating investigation into the influence of physician-involved user-generated content on patient purchase behavior [10,11]. To the best of our knowledge, this is the first investigation into the effects of different user-generated content on user purchase decisions in online healthcare communities. This not only underscores the importance of user-generated content [17,41] but also validates the prominent signals implicit in such content [27,42]. Our findings further affirm that a notable characteristic of online healthcare communities lies in the principles of information dissemination by physicians [43].

*5.3. Practical Implications*

This study provides management insights and service strategies for healthcare service providers, some of which may contribute to increased profitability. First, physicians should ensure the accuracy of displayed content, maintain and update health information regularly, and prevent the dissemination of false or misleading information that could harm their reputations. Second, we suggest that physicians utilize pricing and responsiveness as marketing strategies to enhance performance. To bolster patient trust in healthcare providers, physicians should maintain a high level of responsiveness and establish a positive online presence by demonstrating high engagement with potential clients. It is also advisable for physicians to consider pricing as a significant factor in attracting patient attention and to be more cautious when pricing their medical services. Additionally, to maintain positive user feedback, physicians should actively listen to user suggestions and feedback, make targeted improvements to their services, and consistently provide high-quality medical services.

The findings of this study provide valuable insights for market operators to activate the online healthcare market and enhance the competitiveness of the online healthcare community. First, platform-operating agencies should revise reasonable user-generated content policies to encourage physicians to actively share medical knowledge, disclose personal information, and reduce the risk of information asymmetry. For example, incentive measures could be implemented to encourage physicians to regularly maintain their personal profiles. Second, online healthcare communities should strengthen content review mechanisms to screen user-submitted content and prevent the dissemination of misleading or harmful information, thus providing users with a higher-quality medical content environment. Through these efforts, this study contributes to the sustainable development of online healthcare communities, fostering a culture of transparency, reliability, and collaboration within the online healthcare community.

*5.4. Conclusions*

Drawing form the trust theory and signaling theory, we reveal the influence of user-generated content in online healthcare communities on patient purchasing decisions and examines their boundary conditions. Based on the results from 10,000 physicians on a leading online healthcare platform, we expand the literature on online healthcare by analyzing the importance of user-generated content from different sources. Additionally, by uncovering the moderating effects of price, responsiveness, and comment consistency, we provide empirical evidence on the boundary conditions of the relationship between user-generated content and consultation service purchases. These findings reinforce trust transfer theory and signaling theory. More importantly, to ensure the sustainable development of online healthcare communities, we provide invaluable recommendations for service providers and platform operators to optimize service strategies and cultivate a more conducive market environment.

**Author Contributions:** Conceptualization, X.Y. and H.W.; methodology, H.W.; investigation, H.W.; resources, X.Y.; writing—original draft, H.W.; writing—review & editing, X.Y. and Z.C.; supervision, X.Y. All authors have read and agreed to the published version of the manuscript.

**Funding:** This work was supported by the National Natural Science Foundation of China (72072013), Program for Young Excellent Talents, UIBE (21YQ11), and the Scientific Research Laboratory of AI Technology and Applications, University of International Business and Economics(UIBE); Fundamental Research Funds for the Central Universities in UIBE: 14QN03.

**Institutional Review Board Statement:** Not applicable.

**Informed Consent Statement:** Not applicable.

**Data Availability Statement:** Data are contained within the article.

**Acknowledgments:** We thank all the participants who assisted with the research.

**Conflicts of Interest:** The authors declare no conflicts of interest.

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
