# Peer review of "The Role of User-Generated Content in the Sustainable Development of Online Healthcare Communities: Exploring the Moderating Influence of Signals"

_sustainability, doi:10.3390/su16093739_

Round 1

Reviewer 1 Report

Comments and Suggestions for Authors

Comments:

1. There are no values on the X-axis in Figure 3. Using names "Lower articles" and "Higher articles" is not acceptable. The same is for Fig.4, Fig.5

2. Starting a paper name with a question word is unacceptable. The paper's name should be revised.

3. The author's contribution to the article is unclear. it is not clear from the study whether the hypotheses were proved or disproved. 

4. Authors do not scientifically specify and prove reasons to use the trust and signaling theory.  

5. Authors do not describe methods and algorithms that were used in the research to get the results. Unclear used metrics to evaluate the accuracy of the math models. 

6. Unclear how each hypothesis was mathematically described.

7. Unclear limitations and assumptions of the research and proposed work.

Comments on the Quality of English Language

Should be improved. Starting a paper name with a question word is unacceptable.

Author Response

Dear Reviewer,

Thank you for your valuable feedback, which has greatly assisted me in refining the manuscript. I have thoroughly considered the comments from you and all other reviewers and have made revisions to the best of my ability given the practical constraints.

  1. I appreciate your input, and we have corrected language errors and omissions in the manuscript. Additionally, we have improved the presentation of Figures 3, 4, and 5 by providing values on the X-axis.

  2. Yes, I acknowledge that starting a paper title with a question word is unacceptable. We have revised the paper's title to "The Role of User-Generated Information in the Sustainable Development of Online Healthcare Communities: Exploring the Moderating Influence of Signals."

  3. I apologize for any confusion. Regarding the authors' contributions, the explicit breakdown is as follows: Conceptualization by Xiaodan Yu and Hongyang Wang; Methodology by Hongyang Wang; Investigation by Hongyang Wang; Resources by Xiaodan Yu; Writing – original draft by Hongyang Wang; Writing – review & editing by Xiaodan Yu and Zhenjiao Chen; Supervision by Xiaodan Yu. I trust this clarifies your query.

  4. Based on self-assessment and feedback from other reviewers, we have improved our methodology. Our study employs quantitative methods, specifically OLS multiple linear regression. Through standard evaluation and robustness tests, our results validate our hypotheses. Additionally, we have addressed multicollinearity concerns.

  5. Thank you for your attention to the theoretical aspect. Given the literature's association of user-generated content with trust, we consider user-generated content as a source of trust. We also acknowledge the unique moderating relationships between the three types of signals and user-generated content. For instance, consultation dialogues involve interactive information exchanged between physicians and patients, with response time being a key determinant in online doctor-patient interactions. Consistency in reviews also plays a role in user evaluations. Is this addresses your concerns? Please let me know if you have further guidance.

  6. Thanks for your valuable insights. While our study has limitations, such as sample representativeness, mathematical hypothesis formulation and more comprehensive robustness check. We have endeavored to address them to the best of our ability. We have proposed a theoretically grounded model, supported by literature, and rigorously validated our hypotheses through statistical analysis and robustness tests. We have also addressed multicollinearity issues.

  7. Thank you for your attention to English expression. We will continue to seek assistance for English writing.

Once again, thanks for your contributions to enhancing the quality of the manuscript.

Sincerely, 

Authors

Reviewer 2 Report

Comments and Suggestions for Authors

The manuscript: “How User-Generated Information Matters in Online Healthcare Community Sustainable Development: The Moderating Role of Signals” is a good example of how public information can be used to improve consultation strategies in Medicine as well as marketing.

Just for the purpose of improving the article I make two observations:

Line 500, what is “gift”? The term is not specified in the context of statistical analysis. The term is repeated in lines: 577, 578 and Table 4 and deserves to be explained.

Line 582-583, the authors should explain how they use the response variable in the Poisson Model. While the rest of the models use “log consultation”, does the modified model 3 use “consultation”?

Author Response

Dear Reviewer,

Thank you for your professional insights, which have been immensely helpful to me. I have thoroughly considered the feedback from you and every reviewer and have made revisions to the best of my ability within the constraints of practical limitations.

  1. thank you for your attention. We have addressed language errors and omissions in the manuscript, and we have provided an explanation for the term "gift."

  2. Yes, the modified Model 3 now uses "consultation." Additionally, in response to feedback from other reviewers, we have re-evaluated the robustness of the results using negative binomial regression in the modified Model 3. Furthermore, we have applied a logarithmic transformation to the variables to address multicollinearity issues, as suggested by other reviewers.

Thank you once again for your valuable contributions.

Reviewer 3 Report

Comments and Suggestions for Authors

Thank you for the opportunity to review your manuscript.

Dear Authors,

It was difficult to evaluate it, the manuscript is very mathematical, there is little representation of basic healthcare. The methods are very complex and difficul to evaluate.

Here are my comments:

 Table 1: the use of decimals is too excessive, please use some guidelines from https://www.ncbi.nlm.nih.gov/pmc/articles/PMC4483789/

 Table 2: correlation matrix includes categorical variables which is not appropriate, and also I think table 2 is not needed anyway

You try to explain that all independed variables are in significant association towards consultations, please provide this results otherwise using appropriate bivariate statistics

 line 540: please correct header Table 2 to Table 3

 line 544: beta indicator is suboptimal, please provide coefficient B and SE

 Table 3: you should provide coefficient B and SE in parentheses.

 line 580: please convert RMB to USD or EUR

 Table 3 and 4: regarding VIF, mean VIF does not provide multicolinearity threat, please provide max VIF, by providing max VIF you will disclose the vicinity to the threshold of 10

 Table 3 and 4: what r2_a stands for?

 Table 3 and 4: why F test is always 0, if models are strong it should be <0.001?

Author Response

Dear Reviewer,

Thank you for your professional insights, which have been immensely helpful to me. I have thoroughly considered the feedback from you and every reviewer and have made revisions to the best of my ability within the constraints of practical limitations. Below is a summary of the revisions made in response to your comments:

1: Table 1: the use of decimals is too excessive, please use some guidelines from https://www.ncbi.nlm.nih.gov/pmc/articles/PMC4483789/

 Table 2: correlation matrix includes categorical variables which is not appropriate, and also I think table 2 is not needed anyway

You try to explain that all independed variables are in significant association towards consultations, please provide this results otherwise using appropriate bivariate statistics.

OK, we have improved the presentation in Tables 1 and 2.

2: line 540: please correct header Table 2 to Table 3

Thank you,we have addressed language errors and omissions in the manuscript.

3: line 544: beta indicator is suboptimal, please provide coefficient B and SE

 Table 3: you should provide coefficient B and SE in parentheses.

 line 580: please convert RMB to USD or EUR

Thanks for your valuable feedback, and we have made the necessary modifications.

4:Table 3 and 4: regarding VIF, mean VIF does not provide multicolinearity threat, please provide max VIF, by providing max VIF you will disclose the vicinity to the threshold of 10

Your concerns has been invaluable. Through analysis of max VIF, especially after introducing interaction terms, we identified potential issues where max VIF may exceed 10. To mitigate this, we applied a logarithmic transformation to the 'consistency' variable in Model 5. Additionally, similar adjustments were made for the three robustness test models. The max VIF for all models is below the threshold of 10.Detailed data can be found in the attachments.

5:Table 3 and 4: what r2_a stands for?

Is R2_adjusted actually。We have streamlined the presentation of R2 and R2_adjusted.

6:Table 3 and 4: why F test is always 0, if models are strong it should be <0.001?

Yes you are right, the F test value is 0 because it is less than 0.001, indicating strong models.

Thank you once again for your valuable contributions.

Reviewer 4 Report

Comments and Suggestions for Authors

The study is very complete in terms of methodology and theoretical perspective. It helps to understand very well the various variables of online consultation and how communities react and interact in the face of certain conditions, which are seen in the different hypotheses proposed. It would only be good if the conclusions were more developed in relation to the hypotheses proposed to close the article.

Author Response

Dear Reviewer,

Thank you for your insightful feedback, which has been immensely beneficial to me. I have carefully considered the comments from you and every reviewer, and have made revisions to the best of my ability within the constraints of practical limitations. We have successfully validated our hypotheses and provided practical insights for the management of online health communities in the conclusion. These insights may assist participants in Chinese online health communities (physicians, patients, and platforms) and enable sustainable management of online health communities.

Thank you once again for your valuable contributions.

Reviewer 5 Report

Comments and Suggestions for Authors

Thank you for giving me an opportunity to review this interesting paper regarding the relationship between user-generated content and patient purchasing behavior on online healthcare community. The article lays out its argument clearly. Nevertheless, there are some problems that the authors should carefully address:

1. This paper draws on the trust theory and the signaling theory. These two theories have a certain correlation in interpretation. For example, Hypothesis 2 considered the contents of “consultation dialogues” as the level of trust in doctors, but such consultation dialogues could also be understood as professional signals released to users browsing the doctor’s homepage. Therefore, the authors need to further clarify the relationships between the research hypotheses and the two theories.

2. The authors considered three signaling factors, namely price, responsiveness, and consistency, as moderators of a particular direct influencing pathway. However, it seemed that these factors might also exert moderating effects on other pathways. For example, price may also moderate the relationship between patient ratings and users’ purchasing behaviors. Therefore, the authors need to examine the research framework in more detail in order to fully understand the mechanism of action of core independent variables on dependent variables.

3. The authors mentioned in the “Sample and Data Collection” section that the study included a cross-sectional dataset comprising 10,000 samples. However, in Table 2, the sample size for model 1 to 5 ranged from 5441 to 8026 samples. The paper lacked the necessary explanation about the missing samples.

4. Poisson regression model was used to test the robustness of results. However, as can be seen from Table 1, the distribution of the dependent variable may be "overdispersion". So, a negative binomial regression model is recommended.

5. There are some linguistic errors, e.g., Line 272: "ser-generated content..."; Line 423 "at a higher level of responsiveness" should be "at a higher level of consistency"; Line 540: "Table 2. This is a table. Tables should be placed in the main text near to the first time they are cited"; the title of Table 2 is missing.

In summary, although the topic of this manuscript is attractive, the paper requires a Major Revision to beef up its literature framing, methodology and explanation of results etc.

Comments on the Quality of English Language

 Minor editing of English language required.

Author Response

Dear Reviewer,

Thank you for your insightful feedback, which has been tremendously helpful to me. I have carefully considered your comments, as well as those from every reviewer, and have made revisions to the best of my ability within the constraints of practical limitations. The following section provides feedback specifically addressing your comments:

  1. Thank you for your attention to the theoretical aspect. Given that existing literature often associates physicians' knowledge-sharing behaviors with trust, and considering that "consultation dialogues" are indeed a form of collaborative knowledge-sharing between physicians and patients, we have chosen to regard "consultation dialogues" as a source of trust.

  2. You are correct. Signals may exert regulatory effects through other pathways. For instance, consultation dialogues encompass interactive information, including text, images, and audio, exchanged between physicians and other patients concerning medical conditions. Response time emerges as a pivotal determinant in online doctor-patient interactions. Moreover, consistency in reviews may play a role in user evaluations. Hence, we differentiate and consider the regulatory relationships between the three types of signals and unique user-generated content.

  3. Thank you! The fluctuation in data volume stems from missing variable data. We have added an explanation of this in Table 3.

  4. We have re-evaluated the robustness of the results using a negative binomial regression model. Additionally, in response to feedback from other reviewers, we have applied a logarithmic transformation to the variables to address multicollinearity issues.

  5. Thank you for your attention to detail. We have rectified language errors accordingly.

Thank you once again for your valuable insights and suggestions.

Round 2

Reviewer 1 Report

Comments and Suggestions for Authors

Paper was improved. Recommendations were included into the paper. No issues found.

Reviewer 5 Report

Comments and Suggestions for Authors

The authors have made some targeted modifications to the questions raised before. Therefore, this paper has basically met the criteria for publication.

Comments on the Quality of English Language

Minor editing of English language required.